# Binding of Amyloid β(1–42)-Calmodulin Complexes to Plasma Membrane Lipid Rafts in Cerebellar Granule Neurons Alters Resting Cytosolic Calcium Homeostasis

**DOI:** 10.3390/ijms22041984

**Published:** 2021-02-17

**Authors:** Joana Poejo, Jairo Salazar, Ana M. Mata, Carlos Gutierrez-Merino

**Affiliations:** 1Instituto de Biomarcadores de Patologías Moleculares, Universidad de Extremadura, 06006 Badajoz, Spain; joanapoejo86@gmail.com (J.P.); jairochemsalazar@gmail.com (J.S.); anam@unex.es (A.M.M.); 2Departamento de Química, Universidad Nacional Autónoma de Nicaragua-León, León 21000, Nicaragua; 3Departamento de Bioquímica y Biología Molecular y Genética, Facultad de Ciencias, Universidad de Extremadura, 06006 Badajoz, Spain

**Keywords:** amyloid β(1–42), calmodulin, lipid rafts, cerebellar granule neurons, cytosolic calcium homeostasis, L-type calcium channels

## Abstract

Lipid rafts are a primary target in studies of amyloid β (Aβ) cytotoxicity in neurons. Exogenous Aβ peptides bind to lipid rafts, which in turn play a key role in Aβ uptake, leading to the formation of neurotoxic intracellular Aβ aggregates. On the other hand, dysregulation of intracellular calcium homeostasis in neurons has been observed in Alzheimer’s disease (AD). In a previous work, we showed that Aβ(1–42), the prevalent Aβ peptide found in the amyloid plaques of AD patients, binds with high affinity to purified calmodulin (CaM), with a dissociation constant ≈1 nM. In this work, to experimentally assess the Aβ(1–42) binding capacity to intracellular CaM, we used primary cultures of mature cerebellar granule neurons (CGN) as a neuronal model. Our results showed a large complexation of submicromolar concentrations of Aβ(1–42) dimers by CaM in CGN, up to 120 ± 13 picomoles of Aβ(1–42) /2.5 × 10^6^ cells. Using fluorescence microscopy imaging, we showed an extensive co-localization of CaM and Aβ(1–42) in lipid rafts in CGN stained with up to 100 picomoles of Aβ(1–42)-HiLyteTM-Fluor555 monomers. Intracellular Aβ(1–42) concentration in this range was achieved by 2 h incubation of CGN with 2 μM Aβ(1–42), and this treatment lowered the resting cytosolic calcium of mature CGN in partially depolarizing 25 mM potassium medium. We conclude that the primary cause of the resting cytosolic calcium decrease is the inhibition of L-type calcium channels of CGN by Aβ(1–42) dimers, whose activity is inhibited by CaM:Aβ(1–42) complexes bound to lipid rafts.

## 1. Introduction

Lipid rafts are plasma membrane submicrodomains of sizes between 10 and 100 nm enriched in cholesterol and gangliosides [1]. Amyloid β (Aβ) peptides interact with both cholesterol and gangliosides in ganglioside-clustered raft-like membrane microdomains, which potentiate the formation of Aβ oligomers and fibrils in a cholesterol-dependent manner [2,3,4,5,6,7,8]. Therefore, due to the high content of gangliosides in the brain and in particular in neurons, lipid rafts can be seen as primary targets for Aβ peptide cytotoxicity. Indeed, it has been reported that exogenous oligomeric Aβ applied to neurons in culture concentrates in lipid rafts [9]. Noteworthy, dimeric nonfibrillar Aβ has been shown to rapidly accumulate in lipid rafts in the Tg2576 mouse model of Alzheimer’s disease (AD) [10], and several studies have reported that lipid rafts play an active role in extracellular Aβ uptake and internalization in neurons, reviewed in [11]. Aβ(1–42) is the prevalent Aβ peptide found in the amyloid plaques of AD patients [12], and it has been shown that reuptake of extracellular Aβ(1–42) into neurons can lead to the formation of intracellular aggregates, resulting in neuronal damage and neurotoxicity [13,14,15]. It is to be remarked that oligomeric species of Aβ(1–42) are tightly linked to AD pathogenesis and are presumed to be the cause of neuronal damage [16]. This hypothesis is further supported by the results obtained with the triple transgenic (3xTg) AD mice, where the level of intraneuronal Aβ correlates with synaptic dysfunction and memory impairment [17,18].

Dysregulation of intracellular calcium homeostasis in neurons is a common metabolic feature in both sporadic and familial forms of AD, which by itself can account for enhanced intraneuronal Aβ production and tau hyperphosphorylation potentiation [19,20]. Noteworthy, calmodulin (CaM), the major calcium buffering protein present in the neuronal cytosol, is significantly decreased in the brain of AD individuals [21]. In a previous work, we showed that Aβ(1–42) and Aβ(25–35) had very high affinity for purified CaM, with dissociation constants around 1 nM [22]. However, CaM is not only a cytosolic calcium buffering protein, it is also an intracellular calcium signaling messenger that modulates many neuronal functions whose impairment can lead to significant alterations in neuronal metabolism, excitability, and synaptic activity, and eventually leads to neuronal death (see for example [23,24,25]). On these grounds, it has been proposed that impairment of intracellular calcium buffering by Aβ can trigger a pathogenic feed-forward cycle that leads to altered synaptic morphology, neuronal apoptosis, and cognitive impairment [26].

Neuronal lipid rafts are also platforms that have the ability to generate short lived and dynamic high calcium compartments near the plasma membrane of a thickness lower than 1 micrometer [27,28]. In mature cerebellar granule neurons (CGN) in culture, these high calcium sub-microcompartments are built up by the association with lipid rafts of L-type calcium channels (LTCCs), N-methyl D-aspartate receptors, and plasma membrane calcium pumps (PMCA) [27]. As LTCCs and PMCA are the plasma membrane calcium transport systems that play the major role in the control of resting cytosolic calcium concentration in CGN in culture in a 25 mM potassium medium [29,30,31], lipid rafts also play a major role in the modulation of the excitability of CGN. Interestingly, both LTCCs and PMCA are proteins that bind CaM [32,33,34,35,36]. In addition, other CaM binding proteins are also associated with lipid rafts in mature CGN and other neurons such as calmodulin-dependent protein kinase II (CaMKII) [31,37] and the neuronal isoform of nitric oxide synthase (nNOS) [28,38,39,40]. Most of these CaM-binding proteins have been reported to be inhibited by Aβ peptides, namely, CaMKII [41,42], PMCA [36,43], and nNOS [44,45]. However, the possibility of LTCC inhibition by Aβ(1–42) remains to be experimentally assessed, despite the relevance of LTCC activity for resting cytosolic calcium concentration in neurons, in long-term potentiation/depression (LTP/LTD) [46], and in neuronal function, memory, and cognition [47].

Furthermore, it has been noted that several proteins linked to the production of Aβ have putative calmodulin binding domains (CaMBDs), and it has been shown that CaM binds to amyloid β precursor protein and β-secretase (BACE1; beta-site amyloid β precursor protein cleaving enzyme 1) [25,48].

This work was performed with mature CGN in culture with the following major aims: (1) to demonstrate CaM complexation with submicromolar Aβ(1–42) concentrations in neurons; (2) to experimentally assess the co-localization of Aβ(1–42) and Aβ(1–42):CaM complexes in neuronal lipid rafts; (3) to experimentally evaluate the alteration of the resting cytosolic calcium concentration by submicromolar concentrations of internalized Aβ(1–42); and (4) to identify the primary cause of the observed alteration of the resting cytosolic calcium. Upon reaching all these aims, the results of this work revealed that the inhibition of LTCCs by incubation of CGN with Aβ(1–42) dimers for 2 h elicited a large decrease of resting cytosolic calcium in a partially depolarizing 25 mM potassium medium.

## 2. Results

### 2.1. Co-Immunoprecipitation of Calmodulin (CaM) with Submicromolar Concentrations of Aβ(1–42) in Cerebellar Granule Neurons (CGN) Lysates

In a previous work [22], we showed that the dissociation constant of the complex of Aβ(1–42) with purified CaM is around 1 nM. Since CaM is a protein that is highly expressed in neurons, it seems likely that this should be a major protein target for nanomolar concentrations of Aβ(1–42). Since Aβ(1–42) has been shown to bind to lipid rafts (see above), before running the co-immunoprecipitation assays, the CGN lysates were treated with 10 mM methyl-β-cyclodextrin during 30 min to solubilize lipid rafts and allow for dissociation of the proteins associated with these subcellular structures. The formation of CaM:Aβ(1–42) complexes in mature CGN has been demonstrated by the co-immunoprecipitation of CaM with the anti-Aβ(1–42) antibody in CGN lysates in the presence of 0.25 micromoles of Aβ(1–42)/L, performed as described in detail in the Materials and Methods section (Figure 1A). Of note, the precipitation of a small fraction of CaM molecules observed in the absence of Aβ(1–42) is likely due to its association with poorly solubilized membrane fragments, because it has been demonstrated that CaM also binds with high affinity to several proteins that are associated with lipid rafts in mature CGN, like PMCA [27,49], LTCCs [27,40,50], and CaMKII [31,37].

The amount of total CaM per microgram of CGN lysate protein was quantified using western blots with different lanes loaded with known amounts of purified CaM and of CGN lysates (Figure 1B). The analysis of the results yielded on average 5.5 ± 0.5 ng of CaM/µg of CGN protein. Taking into account that the Petri plates of 35 mm diameter seeded with 2.5 × 10^6^ CGN cells used in this work contained about 170 ± 20 µg of CGN protein, we calculated that on average, there was 935 ± 110 ng of CaM per plate (i.e., 56 ± 6 picomoles of CaM per plate). Since the Aβ(1–42) solutions prepared in this work were largely dimers with around 10% of trimers (see Appendix A in the Materials and Methods section), this implies that in these Petri plates containing 2.5·10^6^ CGN cells, up to 120 ± 13 picomoles of Aβ(1–42) can be bound by CaM.

### 2.2. Fluorescence Resonance Energy Transfer (FRET) Imaging Shows an Extensive Co-Localization between CaM and Aβ(1–42) HiLyte^TM^-Fluor555 in Mature CGN

We used fluorescence resonance energy transfer (FRET) imaging to experimentally assess the spatial proximity between Aβ(1–42) and CaM in mature CGN. To this end, we used the fluorescent derivative Aβ(1–42) HiLyteTM Fluor555. In addition, we aimed to perform these measurements with the lowest Aβ(1–42)-HiLyte™-Fluor555 concentrations as possible to highlight only the subcellular location of the high affinity binding sites for Aβ(1–42), and using an excitation filter of 470 nm instead of 556 nm to minimize the background fluorescence arising from the binding of Aβ(1–42)-HiLyte™-Fluor555 to the Petri plate (see above). Furthermore, these experimental conditions will specifically highlight intracellular Aβ(1–42) binding sites close to flavoproteins like nNOS and the isoform 3 of cytochrome b_5_ reductase, which in mature CGN are associated with lipid rafts [27,40,51,52]. Fluorescence microscopy images of fixed CGN stained with different concentrations of Aβ(1–42)-HiLyte™-Fluor555 using an exposure time of 0.4 s (Figure 2) showed that concentrations of Aβ(1–42)-HiLyte™-Fluor555 as low as 50–100 nanomoles/L gave a fluorescence intensity significantly higher than the CGN autofluorescence. To minimize the contribution of the endogenous red autofluorescence of CGN, we selected a concentration of 100 nanomoles/L of Aβ(1–42)-HiLyte™-Fluor555 for the FRET experiments. In addition, these images and others not shown pointed out that part of Aβ(1–42)-HiLyte™-Fluor555 adsorbed non-specifically to the Petri dish plastic. To quantify this unspecific adsorption, 100 picomoles of Aβ(1–42)-HiLyte™-Fluor555 monomers was added to a Petri plate with 1 mL of MLocke’s K25, incubated under mild stirring during 30 min, and then pipetted from the Petri dish to quartz fluorescence cuvettes. Afterward, the fluorescence was measured in a fluorimeter with excitation and emission wavelengths of 550 and 573 nm and compared with the fluorescence intensity of standard solutions of Aβ(1–42)-HiLyte™-Fluor555 prepared directly in the quartz cuvette. The results allowed us to calculate that the free concentration of Aβ(1–42)-HiLyte™-Fluor555 readily available for binding to CGN cells was only half of the total Aβ(1–42)-HiLyte™-Fluor555 added to the Petri plate (i.e., at most 50 picomoles of Aβ(1–42)-HiLyte™-Fluor555 monomers in the total 1 mL volume of each plate).

FRET imaging between anti-CaM conjugated with IgG-Alexa 488 (anti-CaM*A488) was used to experimentally confirm the formation of CaM:Aβ(1–42) complexes and co-localization of Aβ(1–42) with the CaM expressed in mature CGN. Representative fluorescence microscopy images of mature CGN fixed and stained with anti-CaM*A488 in the absence and presence of 100 picomoles of Aβ(1–42)-HiLyte™-Fluor555/mL are shown in Figure 3A. The merged image highlighted an extensive co-localization (pixel size 0.2 μm) between anti-CaM*A488 and Aβ(1–42)-HiLyte™-Fluor555, both in the neuronal somas and extensions. The increase in the red/green fluorescence intensity ratio after subtraction of the red intensity by direct excitation of Aβ(1–42)-HiLyte™-Fluor555 (Figure 3B) demonstrated that most of the anti-CaM*A488 molecules lay within an efficient FRET distance range with Aβ(1–42)-HiLyte™-Fluor555 (i.e., <50 nm) for this donor–acceptor pair. Moreover, the red fluorescence intensity of Aβ(1–42)-HiLyte™-Fluor555 was more than 80% quenched by the addition of 5 mM CoCl_2_ to the Petri plate, yielding red fluorescence microscopy images like the CGN autofluorescence image shown in Figure 2. Of note, the inner filter effect due to the absorbance of 5 mM Co^2+^ in the wavelength range 470–550 nm, calculated as indicated in the Materials and Methods section, can only account for less than 5% quenching of this fluorescence. Co^2+^ is a well calcium channel blocker at millimolar concentration, and has also been shown to bind to Ca^2+^ sites in proteins (see for example [53,54]). Co^2+^ has a R_0_ value of 1.2 nm as the FRET acceptor of the green fluorescence of fluorescein [55]. Therefore, due to the small CaM size, our results were fully consistent with the extensive complexation of Aβ(1–42)-HiLyte™-Fluor555 with CaM shown by FRET imaging in CGN.

### 2.3. FRET Imaging Highlights the Association of Aβ(1–42)-HiLyte^TM^-Fluor555 and CaM with Lipid Rafts Markers in Mature CGN

In order to experimentally show that Aβ(1–42)-HiLyte™-Fluor555 extensively co-localizes with protein markers of the lipid rafts of mature CGN, we used antibodies against caveolin-1 (Cav-1) and HRas, protein markers of CGN lipid rafts matured in vitro [27], and also cellular prion protein (PrPc), a neuronal marker of lipid rafts that has been shown to form complexes with Aβ [56,57]. Representative fluorescence microscopy images of mature CGN stained with anti-Cav-1 conjugated with IgG-Alexa 488 (anti-Cav1*A488), anti-HRas conjugated with IgG-Alexa 488 (anti-HRas*A488), and anti-PrPc conjugated with IgG-Alexa488 (anti-PrPc*A488) in the absence and presence of 100 picomoles of Aβ1–42-HiLyte™-Fluor555/mL are shown in the Figure 4A–C. In all these cases, the addition of Aβ(1–42)-HiLyte™-Fluor555 elicited a large attenuation of the green fluorescence and an increase in red fluorescence intensity that was higher than the red fluorescence increase expected for the direct excitation of the added Aβ(1–42)-HiLyte™-Fluor555. The analysis of the red/green fluorescence intensity ratio after correction for the red fluorescence due to direct excitation of Aβ(1–42)-HiLyte™-Fluor555 pointed out extensive FRET between Alexa 488 conjugates of anti-Cav1, anti-HRas, and anti-PrPc and the Fluor555 dye bound to Aβ (Figure 4D). Noteworthy, the donor/acceptor FRET pair that showed a higher increase in the ratio of the red/green fluorescence intensity was anti-PrPc*A488/ Aβ(1–42)-HiLyte™-Fluor555, which is in good agreement with the reported formation of complexes between PrPc and Aβ, and also with the fact that in this work, we did not find co-immunoprecipitation of Cav-1 and of HRas with Aβ (see Appendix A). This confirmed an extensive association of Aβ with lipid raft submicrodomains of mature CGN. The merge images also highlight a much larger density of lipid rafts in neuronal somas, since neuronal extensions were less intensely stained with anti-Cav-1*A488, anti-HRas*A488, and anti-PrPc*A488. This latter result is confirmatory of other FRET results reported in previous works with CGN matured in vitro [27,40].

Representative fluorescence microscopy images of mature CGN stained with anti-CaM conjugated with IgG-Alexa 488 (anti-CaM*A488), anti-HRas conjugated with IgG-Cy3 (anti-HRas*Cy3), and with both anti-CaM*A488 and anti-HRas*Cy3 are shown in Figure 5A. The extensive co-localization of CaM and HRas is shown by the merge image, because pixels that maintain the original green and red colors are hardly seen. The large increase in the red/green ratio of fluorescence intensities in CGN double stained with anti-CaM*A488 and anti-HRas*Cy3 with respect to CGN stained only with anti-HRas*Cy3 (Figure 5B) pointed out a high efficiency of FRET between Alexa 488 and Cy3. Thus, these results showed that most of the anti-CaM*A488 and anti-HRas*Cy3 bound to fixed CGN were within the FRET distance using this experimental approach (i.e., <80 nm [27,50]).

### 2.4. A Short Time Incubation with Submicromolar Concentrations of Aβ(1–42) Decreases the Resting Cytosolic Calcium Concentration through Inhibition of L-Type Calcium Channels (LTCCs) in Mature CGN

Figure 6 shows that 48 h incubation of mature CGN with up to 5 μM Aβ(1–42) produced, at most, 10–20% loss of cell viability assayed with the 3-(4,5-dimethylthiazol-2-yl)-2,5-diphenyltetrazolium bromide (MTT) method. Therefore, this allowed us to study the effects on cytosolic calcium dysregulation produced by the incubation of CGN for several hours with micromolar concentrations of this peptide. However, a shorter time of incubation with extracellular micromolar concentrations of Aβ(1–42) is needed to study the effect of submicromolar intracellular concentrations of this peptide on resting calcium concentration, because it has been shown that small oligomeric states of this peptide are internalized in neurons with a half-time of 12–14 h [15].

Figure 7 shows that 2–5 h incubation of CGN in MLocke’s K25 medium with 2 μM Aβ(1–42) produced a statistically significant decrease of the Fura-2 ratio 340/380 from 1.1 ± 0.1 to 0.71 ± 0.07 (*p* < 0.05), in other words, from [Ca^2+^]i 160 ± 30 nM to 71 ± 7 nM. However, this treatment did not significantly change the ratio 340/380 measured after the addition of the LTCC blocker 2 μM nifedipine (*p* > 0.05), as shown also in Figure 7. Therefore, Aβ(1–42) reduces the specific contribution of LTCCs to the resting cytosolic calcium concentration in mature CGN in MLocke´s K25 medium. Indeed, the difference between the ratio 340/380 in the absence and presence of nifedipine decreased from 0.64 in CGN non-treated with amyloid β(1–42) to 0.30 and 0.32 in CGN incubated with 2 μM Aβ(1–42) during 2 and 5 h, respectively. Controls run with up to 2 h incubation with 2 µM of scrambled Aβ(1–42) peptide did not produce alterations of the resting cytosolic calcium (data not shown). Thus, these results pointed out that 2 hours of the incubation of mature CGN with Aβ(1–42) were enough to produce about 50% inhibition of LTCC activity.

The concentration of internalized Aβ after 2 h of incubation of mature CGN with 2 µM Aβ added to the extracellular medium was measured using Aβ(1–42)-HiLyte™-Fluor555, as indicated in the Materials and Methods section. Red fluorescence images of CGN acquired with an excitation filter of 556 nm and a dichroic mirror of 580 nm with an emission filter of 590 nm were analyzed with HCImage software. The increase of red fluorescence intensity in neuronal somas after subtraction of the red autofluorescence of the cells (see Appendix A in the Materials and Methods section) was compared with the red fluorescence intensity of the medium supplemented with different concentrations of Aβ(1–42)-HiLyte™-Fluor555. From these results, we calculated that after 2 h of incubation with 2 µM Aβ in the extracellular medium, the intracellular concentration of Aβ was 193 ± 21 nM of Aβ monomers (i.e., high enough to saturate all CaM present in CGNs).

### 2.5. Extensive FRET between LTCCs Labeled with Alexa488 Fluorescent Antibody and Aβ(1–42)-HiLyte^TM^-Fluor555 in Fixed Mature CGN Is Largely Reversed by Addition of Exogenous CaM

In previous works, we have shown an extensive association of LTCCs with lipid rafts in primary cultures of mature CGNs [27,40,50], and this prompted us to experimentally assess the possibility that both LTCCs and Aβ(1–42) are associated with the same lipid rafts in mature CGN at a concentration of 100 picomoles of Aβ(1–42)/mL (e.g., the concentration of Aβ used in other FRET imaging experiments shown in this work). Representative fluorescence microscopy images of mature CGN stained with anti-LTCC subunit α1C conjugated with IgG-Alexa 488 (anti-LTCCs*A488) and 100 picomoles of Aβ(1–42)-HiLyte™-Fluor555/mL are shown in Figure 8A. Images of several fields of the plate were acquired as indicated in the Materials and Methods before and after the addition of Aβ(1–42)-HiLyte™-Fluor555. A direct inspection of images revealed that the green fluorescence was attenuated after the addition of Aβ(1–42)-HiLyte™-Fluor555. The analysis of the red/green fluorescence intensity ratio after correction for the red fluorescence due to direct excitation of Aβ(1–42)-HiLyte™-Fluor555 showed a two-fold increase in this ratio (Figure 8B), pointing out the occurrence of FRET between Alexa 488 conjugates of anti-LTCC and the Fluor555 dye bound to Aβ. Moreover, a detailed inspection of the merge images revealed significant variations of the extent of FRET between LTCCs and Aβ(1–42)-HiLyte™-Fluor555, as indicated by the color palette of different cellular submicrodomains within the neuronal soma and extensions. The more intense yellow/orange-colored pixels highlighted submicrodomains with higher FRET efficiency. It is noteworthy that this leads to a punctate staining appearance of neuronal extensions that is reminiscent of dendritic spines. Therefore, these results indicate an extensive co-localization of LTCCs and Aβ(1–42)-HiLyte™-Fluor555 within the FRET distance in fixed mature CGN, which using this FRET approach was ≤50 nm, as indicated in the Materials and Methods section.

However, our results excluded significant co-immunoprecipitation of LTCCs by Aβ (see Appendix A). Thus, a direct interaction between LTCCs and Aβ cannot account for the observed inhibition of LTCCs by Aβ. Since CaM modulates LTCCs directly [32,33] or indirectly through CaMK activation [31,58,59], we experimentally assessed the possibility that CaM provides the major anchor point for Aβ near LTCCs in lipid rafts of mature CGN. To this end, we acquired fluorescence microscopy images before and after the addition of 100 picomoles of purified CaM to the Petri plate with CGN stained with anti-LTCCs conjugated with IgG-Alexa 488 (anti-LTCCs*A488) and Aβ(1–42)-HiLyte™-Fluor555 in 1 mL PBS (Figure 9). The concentration of Aβ(1–42)-HiLyte™-Fluor555 was lowered to 50 picomoles for two major reasons: (1) to focus on the higher affinity binding sites for Aβ(1–42)-HiLyte™-Fluor555 in CGN, and (2) to get a CaM/Aβ(1–42)-HiLyte™-Fluor555 monomer molar ratio much higher than 1. The representative images shown in Figure 9A point out that CaM largely reverses the intensity of Aβ(1–42)-HiLyte™-Fluor555 red fluorescence staining of CGN in the somas and more extensively in the thicker neuronal extensions connecting aligned neuronal bodies. Moreover, the addition of 5 mM Co^2+^ elicited more than 80% quenching of the total red fluorescence (Figure 9B), yielding a red fluorescence image that was not significantly different to the image of the red autofluorescence of CGN shown in Figure 2.

## 3. Discussion

The results of this work show that at concentrations lower than 0.25 micromoles of Aβ(1–42)/L, the predominant oligomeric species of this peptide were dimers, with approximately 10% contribution of trimers, and that there was a large complexation of Aβ(1–42) with CaM in mature CGN. This was supported by extensive FRET between anti-CaM stained with the Alexa 488 fluorescent secondary antibody and 0.1 micromoles of Aβ(1–42)-HiLyte^TM^-Fluor555/L in fixed and permeabilized CGN, and also by co-immunoprecipitation of CaM with anti-Aβ(1–42) in the presence of 0.25 micromoles of Aβ(1–42)/L. Moreover, taking into account the high content of CaM in neurons ( [60,61] and this work) and the very low dissociation constant of the CaM:Aβ(1–42) complex, approximately 1 nM [22], our results point that CaM is a major binding target for Aβ(1–42) in neurons, not only in mature CGN. As our results also showed that 2.5 x 10^6^ mature CGN (the number of cells seeded per Petri plate in this work) contained 56 ±6 picomoles of CaM, this implies that in these plates, CaM can bind up to 120 ± 13 picomoles of Aβ(1–42) monomers, decreasing the concentration of free Aβ(1–42) to approximately 1 nM. Although Aβ peptides have also been shown to bind with dissociation constants close to 1 nM to PrPc [62] and to glycogen synthase kinase 3α (GSK3α) [63], it is to be noted that all these alternate target proteins were expressed at much lower levels than CaM in neurons. Thus, CaM can be seen in neurons as the major intracellular target protein for Aβ peptides, helping to keep the free intracellular concentration of Aβ peptides in the low nanomolar range. Therefore, brains affected with Alzheimer’s disease, which contain lower CaM levels than normal brains [21], should be expected to suffer a stronger rise in the free intracellular concentration of Aβ peptides upon β-secretase activation, being more prone to undergoing neuronal degeneration. In addition, CaM:Aβ(1–42) complexes can be by themselves intracellular transducers for focalized actions of Aβ peptides, because CaM has a major role in neuronal metabolism, excitability, and signaling through direct interaction with proteins that play a major regulatory role of neuronal functions in different subcellular locations.

Aβ peptides have been found in neuronal lipid rafts, where they promote specific signaling alterations [11,64,65]. Aβ peptides bind to cholesterol and gangliosides, highly enriched in the lipid rafts of the plasma membrane, and also to phosphatidylserine, a lipid enriched in the neuronal plasma membrane. In addition, PrPc, which is constitutively present in lipid rafts, has been shown to bind Aβ peptides [56,57,62], providing further direct interaction points for the physical anchorage of Aβ to lipid rafts. Using Aβ(1–42)-HiLyte^TM^-Fluor555, we experimentally assessed with FRET imaging that in fixed and permeabilized CGN, Aβ(1–42) binding sites were within the FRET distance from protein markers of plasma membrane lipid rafts stained with specific antibodies. The donor/acceptor FRET pair with higher increase in the red fluorescence/green fluorescence ratio was anti-PrPc stained with the Alexa 488 fluorescent secondary antibody/Aβ(1–42)-HiLyte^TM^-Fluor555, indicating a more extensive co-localization of PrPc and Aβ(1–42) within FRET distance. This is in good agreement with the formation of PrPc:Aβ complexes reported elsewhere (see above). To minimize the contribution of low affinity intracellular binding sites for Aβ, these results were obtained with the addition of only 100 picomoles of Aβ(1–42)-HiLyte^TM^-Fluor555 monomers to a Petri plate containing 2.5 x 10^6^ cells in 1 mL PBS. As the FRET distance limit using this experimental approach was less than 60 nm, these results highlight Aβ binding sites within the same lipid rafts or in close proximity to them. Moreover, fluorescence microscopy images highlighted that there was a large co-localization in the neuronal soma and also in neuronal extensions of Aβ(1–42)-HiLyte^TM^-Fluor555 and protein markers of lipid rafts stained with specific antibodies labelled with fluorescent Alexa 488 secondary antibodies. However, the raft markers Cav-1 and HRas were not co-immunoprecipitated by the anti-Aβ(1–42) antibody in the presence of 0.25 µM Aβ(1–42), pointing out that neither Cav-1 nor HRas provide direct anchoring points of Aβ(1–42) in lipid rafts. On the other hand, FRET imaging with anti-CaM stained with the Alexa 488 fluorescent secondary antibody (as FRET donor) and anti-HRas stained with the Cy3 fluorescent secondary antibody (as FRET acceptor) pointed out that a significant fraction of CaM were separated by less than 80 nm (i.e., at a distance lower than the maximum 100 nm size reported for lipid rafts) [1]. This is consistent with the known association of several CaM-binding proteins with lipid rafts in mature CGN, namely, nNOS, LTCCs, PMCA, and CaMKII [27,31,40].

Previous works have shown that LTCCs associated with lipid rafts play a major role in the control of resting cytosolic calcium homeostasis in mature CGN in vitro in the optimum survival medium (i.e., a partially depolarizing medium containing 25 mM KCl) [30,31]. It has also been reported that LTCC activity is stimulated by phosphorylation mediated by CaMKII [31,58,59], but CaM binding to the LTCC α1C subunit produces inactivation of these channels [32,33]. In this work, we showed that incubation of CGN with 2 µM (monomers) Aβ(1–42) during 2 h, which allowed us to reach 193 ± 21 nM (monomers) of internalized Aβ(1–42) in CGN, produced approximately 50% inhibition of the activity of LTCCs, and that incubation up to 5 h did not significantly increase this inhibition. Since the dimer is the predominant aggregation state of the solutions of Aβ(1–42) used in this work, this means that less than 100 nM Aβ(1–42) dimers inhibited by 50% the LTCC’s activity. Note that there was not a significant change in the resting cytosolic calcium concentration of CGN measured once LTCCs were blocked with nifedipine after these treatments with Aβ(1–42). This result indicates that LTCC is the calcium transport system involved in the control of resting cytosolic calcium in mature CGN in vitro that is most sensitive to Aβ(1–42) under our experimental conditions. Additionally, this decrease in resting cytosolic calcium elicited by Aβ(1–42) should attenuate CGN excitability because of the strong dependence of synaptic activity upon cytosolic calcium concentration.

A remarkable observation is that upon blockade of LTCCs with nifedipine, the resting cytosolic calcium of CGN was not significantly altered by Aβ(1–42). Thus, the formation of calcium pores by intracellular Aβ is unlikely at the short times of exposure to the low nanomolar free intracellular concentrations of Aβ attained in this work. This also implies that the activity of other calcium transport systems involved in the control of cytosolic calcium homeostasis in CGN are not significantly impaired by the treatment with Aβ(1–42) that is enough to elicit inhibition of LTCCs. The major transport system for calcium homeostasis in neurons, which extrudes cytosolic calcium toward the extracellular medium is the PMCA [35]. Despite that Aβ(1–42) can bind and partially inhibit PMCA [36,43], it is to be noted that only a weak inhibition of lipid rafts associated with PMCA by intracellular nanomolar concentrations of Aβ(1–42) should be expected in a calcium concentration range below 0.2 μM [43], the resting cytosolic concentration range measured in mature CGN in this work and in previous works [27,30,31]. Moreover, previous works have shown that the inhibition of PMCA by Aβ(1–42) can be antagonized by CaM and cholesterol [43,66]. Furthermore, it is to be noted that the sustained lowering of cytosolic calcium of CGN in a partially depolarizing medium is a cellular stress observed at short times of exposure to low nanomolar free intracellular concentrations of Aβ. It is likely that at higher concentrations of Aβ or at longer times of exposure to Aβ, this may trigger calcium release from intracellular stores, mainly from endoplasmic reticulum, as a compensatory or adaptive cellular response. Indeed, enhanced endoplasmic reticulum calcium release has been shown in AD [67,68]. Moreover, it has been shown that familial AD mutations of presenilins potentiate endoplasmic reticulum calcium leak and cause early-onset inherited AD [69,70].

However, LTCCs are not co-immunoprecipitated by anti-Aβ(1–42), and this excludes that the inhibition of LTCC activity can be due to the blockade of these calcium channels by direct binding of Aβ(1–42). However, fluorescence microscopy revealed efficient FRET between anti-LTCC stained with Alexa 488 fluorescent secondary antibody and Aβ(1–42)-HiLyteTM-Fluor555. Therefore, Aβ(1–42) must bind to a modulatory site in the lipid rafts located very close to the LTCC structure (i.e., less than few nanometers distance from LTCC subunits since the overall size of the complex between the anti-LTCC primary antibody and the Alexa 488 fluorescent secondary antibody is approximately 40 nm) [27,50]. Since most of this internalized Aβ(1–42) should be bound to CaM, the inhibition of LTCC by Aβ(1–42) can be rationalized in terms of the known inhibition of CaMKII by Aβ peptides [41,42], and also in terms of potentiation by Aβ(1–42) of the inactivation of LTCC by direct interaction with CaM. The hypothesis that CaM:Aβ(1–42) complexes are involved in the modulation of LTCC was experimentally supported by a large reversion by exogenously added CaM of the co-localization between anti-LTCC stained with the Alexa 488 fluorescent secondary antibody and Aβ(1–42)-HiLyte^TM^-Fluor555. Additionally, the large quenching of Aβ(1–42)-HiLyte^TM^-Fluor555 fluorescence by 5 mM Co^2+^ indicated that the binding sites of Aβ(1–42)-HiLyte^TM^-Fluor555 in fixed and permeabilized CGN were separated by less than 2 nm from high affinity calcium sites, as expected for Aβ(1–42)-HiLyte^TM^-Fluor555:CaM complexes [22]. Noteworthy, Aβ has been shown to prevent CaMKII activation in rat hippocampal slices [41], and impairment of the phosphorylation of the β-subunit of LTCC by CaMK inhibitors leads to the strong inhibition of LTCC in partial depolarizing MLocke’s K25 mM medium [31]. In addition, it has been suggested that CaMKII dysregulation may be a modulator of toxicity in Alzheimer’s disease [42]. However, we cannot exclude that Aβ(1–42) association with CaM may also potentiate the inactivation of LTCC activity by direct binding of CaM, a possibility which to the best of our knowledge has been overlooked until now. However, a critical evaluation of this point requires further extensive experimental studies that are out of the scope of this work.

In summary, by using mature CGN in culture, we showed a large complexation of CaM by submicromolar concentrations of Aβ(1–42) dimers, and also extensive co-localization of CaM and Aβ(1–42) within the same lipid rafts in plates seeded with 2.5 × 10^6^ cells and stained with up to 100 picomoles of Aβ(1–42)-HiLyte^TM^-Fluor555 monomers. In addition, we found that the resting cytosolic calcium of mature CGN in partially depolarizing 25 mM potassium medium was largely lowered by exposure to Aβ(1–42) dimers during 2 h, conditions that produced an internalization of less than 100 picomoles of Aβ(1–42) dimers in neuronal somas. As a final conclusion, we identified that the primary cause of this decrease of the resting cytosolic calcium was the inhibition of LTCCs of CGN by Aβ(1–42) dimers.

## 4. Materials and Methods

### 4.1. Preparation of Rat Cerebellar Granule Neurons (CGN)

CGNs were obtained from dissociated cerebella of 7-day-old Wistar rats as described previously [27,29,31,40,51,52,71,72]. Animal handlings were performed in accordance with Spanish regulations and were approved by the Ethical Committee of the University of Extremadura. Briefly, cells were plated in Dulbecco’s Modified Eagle medium (DMEM) supplemented with 10% heat-inactivated fetal bovine serum, 5 mM glucose, 19.63 mM KCl, 3.7 ng/mL insulin, 7 µM 4-aminobenzoic acid, 50 U/mL penicillin, 25 U/mL streptomycin, 0.91 mM pyruvate, and 2 mM glutamine on 35 mm diameter dishes (Corning, NY, USA) coated with poly-D-lysine at a density of 2.5 × 10^6^ cells/dish. CGN cultures were kept at 37 °C in a humidified atmosphere of 95% air/5% CO_2_. Cytosine arabinofuranoside (10 µM) was added to fresh culture medium 48 h after plating to prevent the replication of non-neuronal cells. Seven days after plating, the culture medium was replaced with the serum-free DMEM:F12 medium (1:1) supplemented with 12.5 mM glucose, 20.82 mM KCl, 5 µg/mL insulin, 0.1 mg/mL apo-transferrin, 20 nM progesterone, 50 U/mL penicillin, 25 U/mL streptomycin, 0.1 mg/mL pyruvate, and 2 mM L-glutamine. Mature CGNs at 8–10 days in vitro (DIV) were used in all of the experiments.

The composition of MLocke’s K25 buffer (pH 7.4 at 37 °C) used in this work was as follows: 4 mM NaHCO_3_, 10 mM Tricine, 5 mM glucose, 2.3 mM CaCl_2_, 1 mM MgCl_2_, and 134 mM NaCl/25 mM KCl.

### 4.2. Aβ(1–42) Solutions and Aggregation State

Aβ(1–42) solutions were prepared dissolving the solid lyophilized peptide in 1% NH_4_OH, and thereafter diluted with PBS to the desired concentration, as in [22]. The aggregation state of Aβ(1–42) stock solutions used in this work were evaluated using the rapid photoinduced cross-linking of unmodified proteins approach described in [73,74]. Briefly, Aβ(1–42) diluted to 177 μM in 60 mM NaOH and 10 mM phosphate at pH 7 was sonicated in a water bath, Selecta Ultrasons, set at the maximum intensity of 150 watts during 1 min and, thereafter, it was centrifuged at 16,000 *g* during 10 min. The supernatants were treated with 60 μM Tris(2,2′-bipyridyl)dichloro-ruthenium(II) hexahydrate and 4.4 mM ammonium persulfate, placed in the sample compartment of a Perkin-Elmer 650-40 fluorimeter (Perkin-Elmer, Waltham, MA, USA) and irradiated with the light of 452 nm emitted by the fluorimeter 150 watts Xenon lamp during 1 s at room temperature. Immediately after irradiation, the reaction was stopped by the addition of 5% *v*/*v* β-mercaptoethanol plus sample buffer [95 mM tris-(hydroxymethyl) aminomethane (Tris)-HCl buffer, pH 6.8/3% sodium dodecyl sulfate (SDS)/13% glycerol, and 0.005% bromophenol blue], and heated at 37 °C for 15 min, before loading into the gel for Tricine-SDS-polyacrylamide electrophoresis (PAGE) prepared and run as indicated by [75]. The results obtained (Appendix A) showed a major band of a molecular weight close to 9 kDa and a faint band of molecular weight between 12 and 15 kDa. As the monomer molecular weight of Aβ(1–42) is approximately 4.5 kDa, these results indicate that dimers were the predominant aggregation state of Aβ(1–42) in the stock solutions used in this work, with a minor (<10%) contribution of trimers. Aβ(1–42) monomers were not detected in our stock solutions.

### 4.3. Cell Viability

Cells were incubated with different concentrations of Aβ(1–42) (0.75, 1.5, 2.5 and 5 µM) in serum-free DMEM:F12 medium at 37 °C and 5% CO_2_. After 48 h incubation, CGN plates were washed with 1 mL MLocke’s K25 buffer and cell viability was experimentally assessed measuring the amount of colored formazan by the reduction of 3-(4,5-dimethylthiazol-2-yl)-2,5-diphenyltetrazolium bromide (MTT) as in previous works [27,29,31,52,71,72]. Untreated cells were regarded as controls (100% cell survival) and the cell survival ratio was expressed as the percentage of the control.

### 4.4. Co-Immunoprecipitation

CGN was lysed in buffer 25 mM Tris–HCl, pH 7.4, 150 mM NaCl, 5 mM ethylenediaminetetraacetic acid, 50 mM NaF, 5 mM NaVO_3_, and 0.25% 4-(1,1,3,3-tetramethyl butyl)phenyl-polyethylene glycol (Triton X-100), supplemented with 1x SIGMAFAST^TM^ protease inhibitor cocktail (Sigma-Aldrich, St. Louis, MO, USA). Lysates were collected and supplemented with 50% glycerol. The protein concentration of cell lysates was measured by the Bradford’s method using bovine serum albumin (BSA) as the standard.

Co-immunoprecipitation was carried out using the protein A/G PLUS-Agarose sc-2003 of Santa Cruz Biotechnology Inc. (Santa Cruz, CA, USA) following the instructions given in their technical data sheets. Prior to the co-immunoprecipitation experiment, cell lysates were treated with 10 mM of methyl-β-cyclodextrin for 30 min at 4 °C in a tube-rotor with continuous shaking. Afterward, in an Eppendorf tube, 200 µg of CGN lysate was incubated with 5 µg mouse anti-β-amyloid antibody (Sigma, A8354) at 4 °C with continuous shaking. After 1 h, 65 µL protein A/G PLUS-Agarose was added and incubated overnight at 4 °C with continuous shaking. The next day, the phosphate buffered saline (PBS) control sample or 0.25 µM Aβ(1–42) in PBS (treated sample) was added and incubated for 1 h at 4 °C with continuous shaking. The matrix was precipitated by centrifugation at 2500× *g* during 5 min at 4 °C in a refrigerated Eppendorf microcentrifuge 5415R. The supernatant was removed, and the precipitated matrix was subjected to three washes with 50 µL PBS (control sample) or 50 µL PBS plus 0.25 µM Aβ(1–42) (treated sample). A centrifugation step (2500× *g*, 5 min at 4 °C) was performed in a refrigerated Eppendorf in each washing step. The supernatant was carefully removed and the matrix precipitate was resuspended in 80 µL of electrophoresis sample buffer, boiled during 3 min, and stored at −20 °C until running on an SDS-PAGE gel for western blotting analysis.

### 4.5. Western Blotting

SDS-PAGE was run at concentrations of 7.5%, 10.4%, 13.5%, or 15% acrylamide depending upon the molecular weights of the target proteins using 20 µg CGN lysates or 15–20 µL co-immunoprecipitated sample per lane. Gels were transferred to polyvinylidene difluoride (PVDF) membranes of 0.2 μm average pore size, and PVDF membranes were blocked with 3% BSA in Tris-buffered saline (TBS) supplemented with 0.05% polyoxyethylene sorbitan monolaurate (TBST) for 1 h at room temperature. Before incubation with the primary antibody, membranes were washed six times with TBST. Immunodetection of proteins was performed with their specific antibodies at a dilution of 1:1000 in TBST for anti-caveolin-1, anti-HRas, and anti-LTCC antibodies, and at a dilution of 1:2000 in TBST for anti-CaM and anti-cellular prion protein (PrPc) antibodies. After incubation with the first antibody overnight at 4 °C, membranes were washed six times with TBST and incubated for 1 h at room temperature with the appropriate secondary IgG antibody conjugated with horseradish peroxidase at a dilution of 1:5000 in TBST. Then, membranes were washed six times with TBST followed by incubation for 3 min with the Bio-Rad Clarity Western ECL substrate. Western blots were revealed with Bio-Rad ChemiDocTM XRS+ (Bio-Rad, Hercules, CA, USA) and data analyzed with Image Lab 6.0.1 software.

### 4.6. Measurements of Internalization of Aβ(1–42)-HiLyte^TM^-Fluor555 in Mature CGN

The internalization of Aβ(1–42) was monitored by fluorescence microscopy with Aβ(1–42)-HiLyte^TM^-Fluor555 following an experimental approach similar to that used with other fluorescent derivatives of Aβ(1–42) in cell cultures [15,76]. The extent of internalization of Aβ(1–42) was measured from the increase in red fluorescence in CGN soma at different times of incubation (up to 2 h) at 37 °C and 5% CO_2_ with a total concentration of 2 μM of Aβ(1–42) monomers (1.8 μM of Aβ(1–42) monomers plus 0.2 μM of Aβ(1–42)-HiLyte^TM^-Fluor555 monomers) added to the serum-free DMEM:F12 medium supplemented as indicated above. After the selected time of incubation, the medium was replaced by MLocke’s K25 and the Petri plate was placed in the holder of the fluorescence microscope thermostated at 37 °C for image acquisition. Fluorescence images were acquired with a Hamamatsu Orca-R^2^ CCD camera (binning mode 2 × 2) camera (Hamamatsu, Hamamatsu-city, Japan) attached to a Nikon Diaphot 300 epifluorescence microscope (Tokyo, Japan) with a NCF Plan ELWD 40× objective, using an excitation filter of 556 nm and a dichroic mirror of 580 nm with an emission filter of 590 nm, and 0.03 s exposure time. Quantitative analysis of the average fluorescence intensity per pixel of selected neuronal soma was done with HCImage software using the region of interest (ROI) tool. The results obtained after 2 h incubation are shown in panel A of Appendix A. In separate experiments, the average intensity per pixel obtained with increasing concentrations of Aβ(1–42)-HiLyte^TM^-Fluor555 in Mlocke’s K25 in the Petri plate was recorded for calibration of the Aβ(1–42)-HiLyte^TM^-Fluor555 fluorescence under the same experimental conditions, and the results are shown in panel B of Appendix A. The average intensity readings per pixel in CGN neuronal somas were taken from several fields for a total number of 102 cells, and after subtraction of CGN autofluorescence, yielded an internalized concentration of 19.3 ± 2.1 nM of Aβ(1–42)-HiLyte^TM^-Fluor555 monomers. Assuming that there was not a significant difference between the rate of internalization of Aβ(1–42)-HiLyte^TM^-Fluor555 and of Aβ(1–42), we calculated an internalized concentration of 193 ± 21 nM of total Aβ monomers by interpolation in the calibration line shown in panel B.

### 4.7. Fluorescence Resonance Energy Transfer (FRET) Imaging

FRET imaging was performed as in previous works [27,40,50,51,52]. Fluorescence microscopy images of CGN were acquired with a Hamamatsu Orca-R^2^ CCD camera (Hamamatsu) attached to a Nikon Diaphot 300 epifluorescence microscope (Tokyo, Japan) with a NCF Plan ELWD 40× objective (pixel size of the images shown in this work 0.2 μm). Quantitative analysis of the average fluorescence intensity per pixel of selected neuronal soma was done with the HCImage software using the ROI tool, as in previous works. Only fields devoid of large aggregates of neurons forming granules or small grain-like structures were selected for image acquisition to minimize the distortion of images by the sum of the fluorescence contributions of juxtaposed layers of neurons. The mean ± s.e. intensity reading of fluorescence per pixel within CGN somas were obtained in experiments performed in triplicate (*n* > 100 CGN somas in each case). Images of CGN were acquired with an excitation filter of 470 nm, and 510 nm dichroic mirror/520 nm emission filter (green fluorescence), using the exposure times indicated for each case in the legends for the figures. Untreated cells were regarded as controls.

CGN were washed with MLocke’s K25 buffer to remove the phenol red remaining in the plates. Then, CGN were fixed with 2.5% para-formaldehyde, 3 mM MgCl_2_, 2 mM ethylene glycol-bis(2-aminoethylether)-*N*,*N*,*N*,*N*′-tetraacetic acid, and 0.32 M sucrose in PBS (5 mM sodium phosphate, 137 mM NaCl, and 27 mM KCl, pH 7).

The following selected protein targets for FRET imaging were used: CaM, HRas, caveolin-1 (Cav-1), PrPc, and α1C subunit of LTCC (Cav1.2). Before their use for fluorescence microscopy image acquisition, the specificity of primary antibodies for the selected target proteins was routinely assessed by the presence of the major and most intense band at the expected protein molecular weight in the western blotting run with CGN lysates. In most cases, this band accounts for more than 90% of the total band staining (Appendix A). Primary antibodies used in these experiments were: rabbit anti-CaM (Epitomics, 1716-1, 1:200), goat anti-HRas (Santa Cruz Biotechnology, sc-32026, 1:100), rabbit anti-caveolin-1 (Santa Cruz Biotechnology, sc-894, 1:100), mouse anti-PrPc (Thermo Fisher, 6H4-7500997, 1:100), and rabbit anti-LTCC α1C subunit (Santa Cruz Biotechnology, sc-25686, 1:50).

After CGN were fixed, cells were blocked with 1% BSA in PBS supplemented with 0.2% Triton X-100 (PBST) for 1 h at 37 °C and then incubated during 1 h at 37 °C with the target primary antibody in PBS and washed tree times with PBS (washing step). Thereafter, CGN were incubated for 1 h with the appropriate Alexa488-labeled secondary antibody in PBST (1:200) and washed again three times with PBS before acquisition of fluorescence microscopy images stained only with the donor dye. After finishing the acquisition of FRET donor images, CGN were incubated for 60 min at 37 °C with 50 or 100 nM Aβ(1–42)-HiLyte^TM^-Fluor 555 in PBS with continuous and gentle mixing. Then, the acquisition of fluorescence microscopy images of selected fields, routinely 5–6 fields of each Petri plate, were performed within the next 20–30 min. Contribution of CGN autofluorescence and secondary Alexa488-antibody in the absence of the primary antibody were assessed before running FRET experiments and were found to be lower than 10% of the average fluorescence intensity per pixel obtained with specific primary antibodies. This background signal was subtracted for calculations of the RF/GF ratio obtained with CGN plates. Due to the close absorbance spectrum of Cy3 and Aβ(1–42)-HiLyte^TM^-Fluor 555, significant FRET efficiency using this experimental approach implies that the selected protein was separated by less than 50 nm from Aβ(1–42)-HiLyte^TM^-Fluor 555, as discussed in detail in previous works [27,50].

In the case of FRET imaging using a fluorescent acceptor antibody bound to another protein target (e.g., CaM stained with a Alexa488 fluorescent secondary antibody as donor and HRas stained with a Cy3 fluorescent secondary antibody as acceptor), we followed the protocol described in detail in previous publications of our laboratory [27,50]. Briefly, after finishing the acquisition of FRET donor images, CGN were incubated for 1 h at 37 °C with the acceptor target primary antibody as indicated above, washed three times with PBS, then incubated for 1 h with the appropriate Cy3-labeled secondary antibody in PBST (1:200), and washed again three times with PBS before the acquisition of fluorescence microscopy images. Controls were also run in the absence of the second primary antibody to correct for partial loss of the donor fluorescence produced by the treatment to label the second protein, on average between 20 and 25% loss of the donor fluorescence, and also to subtract background signals.

The average intensity of fluorescence per pixel within CGN somas were taken using the ROI tool of the Hamamatsu HC Image software to select somas as the areas of interest in experiments performed in triplicate, at least, *n* > 100 CGN somas in each case. As discussed in previous works [27,50], significant FRET efficiency using this experimental approach implies that the selected proteins were separated by ≤80 nm.

### 4.8. Measurement of the Intracellular Free Ca^2+^ Concentration ([Ca^2+^]i)

[Ca^2+^]i was measured as indicated in detail in previous works [27,29,31,71]. Briefly, cells were incubated with 2 µM Aβ(1–42) in DMEM:F12 medium for 2 h or 5 h at 37 °C and 5%CO_2_ on a sunflower mini-shaker (BioSan, Labnet, Madrid, Spain) with continuous and gentle mixing. One hour before the end of the incubation with Aβ(1–42), CGN was loaded with 5 μM Fura-2-acetoxymethyl ester (Fura2-AM) and 0.025% Pluronic-F127 at 37 °C. Next, CGN plates were washed twice with 1 mL MLocke’s K25 buffer and the culture dishes were placed in the thermostatic controlled plate (Warner Instrument Co., Hamden, CT, USA) of the Nikon Diaphot 300 inverted epifluorescence microscope. Nifedipine (10 µM) was used to block the L-type calcium channels. Digital images with 340 and 380 nm excitation filters and a 510 nm dichroic mirror/520 nm emission filter were taken with a Hamamatsu Orca-R^2^ CCD camera (binning mode 2 × 2) and Lambda 10–2 filter wheel controller and subsequently analyzed with HCImage software. Data acquisition and analysis were done after the selection of the neuronal soma using the ROI tool of this software. The 340/380 ratio data given in this work were population averages ± s.e. intensity reading of fluorescence per pixel within CGN somas were obtained using the ROI tool of the Hamamatsu HCImage software to select somas as areas of interest in experiments performed in triplicate (*n* > 100 CGN somas in each case).

### 4.9. Chemicals and Reagents

Human Aβ(1–42)-HiLyte™-Fluor555 was obtained from AnaSpec (Freemont, CA, USA). Unlabeled Aβ(1–42) and scrambled Aβ(1–42) were synthesized and supplied by StabVida (Caparica, Portugal). Purified bovine brain CaM was purchased from Sigma-Aldrich (Madrid, Spain).

Primary antibodies: goat anti-HRas (sc-32026), rabbit anti-Cav-1 (sc-894), rabbit anti-LTCC α1C subunit (sc-25686), and rabbit anti-LTCC β subunit (sc-25689) antibodies were supplied by Santa Cruz Biotechnology (Santa Cruz, CA, USA). Rabbit anti-CaM (Epitomics 1716-1) antibody was supplied by Abcam (Cambridge, UK) and mouse anti-PrPc antibody (Thermo Fisher, 6H4-7500997) was purchased from Thermo Fisher Scientific (Madrid, Spain). Monoclonal mouse anti-Aβ antibody (A8354) was purchased from Sigma-Aldrich (Madrid, Spain). Fluorescent-labeled secondary antibodies used to label the primary antibodies listed above were anti-rabbit IgG-Alexa488 (cat. no. A11008), anti-goat IgG-Alexa488 (cat. no. A11055), and anti-mouse IgG-Alexa488 (cat. no. A11001) from Invitrogen (Molecular Probes, Eugene, OR, USA). Anti-rabbit IgG-horseradish peroxidase, anti-goat IgG-horseradish peroxidase, and anti-mouse IgG-horseradish peroxidase were supplied by Sigma-Aldrich. Bio-Rad Clarity Western ECL substrate was purchased from Bio-Rad (Alcobendas - Madrid, Spain).

Fura-2 acetoxymethyl ester and pluronic F-127 were obtained from Biotium (Hayward, CA, USA) and Invitrogen, respectively. Nifedipine was supplied by Sigma-Aldrich (Madrid, Spain). Protein A/G PLUS-Agarose sc-2003 was purchased from Santa Cruz Biotechnology Inc. (Santa Cruz, CA, USA).

All other reagents and chemicals were of analytical grade from Sigma-Aldrich (Madrid, Spain) or Roche–Merck (Darmstadt, Germany).

### 4.10. Statistical Analysis

Results were expressed as the mean standard error (s.e.). Statistical analysis was carried out by the Student’s *t*-test. A significant difference was accepted at the *p* < 0.05 level. All results were confirmed with duplicate measurements of at least three different CGN preparations.

## Figures and Tables

**Figure 1 ijms-22-01984-f001:**
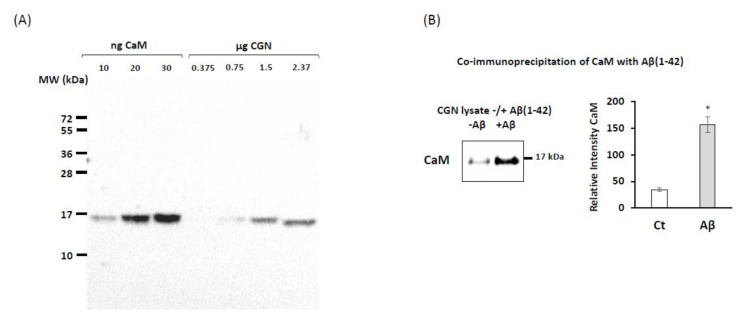
Co-immunoprecipitation between calmodulin (CaM) and Aβ(1–42). (**A**) Quantification of calmodulin content in cerebellar granule neurons (CGN) lysates. (**B**) Western blotting of CaM (anti-CaM 1716-1, Epitomics) after co-immunoprecipitation assay with mouse anti-β-amyloid antibody (Sigma, A8354), as described in the Materials and Methods section.

**Figure 2 ijms-22-01984-f002:**
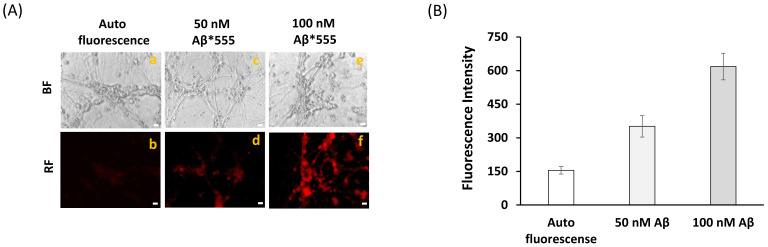
**Fluorescence microscopy images of CGN stained with Aβ(1–42) HiLyte^TM^-Fluor 555.** (**A**) Representative fluorescence microscopy images of mature CGN without staining-autofluorescence (b) and CGN stained with 50 nM (d) or 100 nM (f) of Aβ(1–42) HiLyte^TM^-Fluor 555 incubated for 1 h at 37 °C and 5% CO_2_. Bright field (BF) and red fluorescence (RF) images are shown for representative selected fields. The exposure time for RF images was 0.4 s. Scale bar inserted in fluorescence microscopy images = 10 μm. (**B**) Quantitative fluorimetric analysis of Aβ(1–42) HiLyte^TM^-Fluor 555 bound to CGN. The results yielded a 2.3-fold and 4-fold increase (mean of triplicate experiment ± s.e.) of the fluorescence of CGN stained with 50 nM and 100 nM Aβ(1–42) HiLyte^TM^- Fluor 555 with respect to the CGN autofluorescence, respectively.

**Figure 3 ijms-22-01984-f003:**
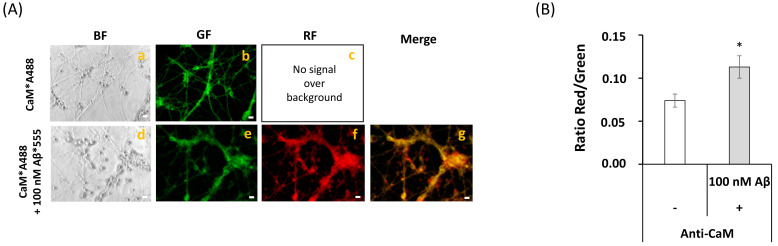
**Extensive FRET between anti-CaM antibody stained with a secondary Alexa488 fluorescent antibody and Aβ(1–42)-HiLyte^TM^-Fluor 555 in fixed and permeabilized mature CGN**. (**A**) Representative quantitative fluorescence microscopy images of CGN stained with anti-CaM antibody (1716-1)/ IgG-Alexa488 (CaM*A488, a–c) or with anti-CaM/IgG-Alexa488 and Aβ(1–42) HiLyte^TM^-Fluor 555 (CaM*A488/ Aβ(1–42), d–g). Bright-field (BF), green fluorescence (GF), and red fluorescence (RF) images are shown for each one of the selected fields, and the orange-yellow areas (Merge image) point out the higher intensity FRET regions (g). The exposure time for GF and RF images was 0.03 s. Scale bar inserted in fluorescence microscopy images = 10 μm. (B) Ratio of red/green fluorescence intensity per pixel (RF/GF) obtained from the analysis of the fluorescence intensity data of CGN somas stained with anti-CaM*IgG-Alexa488 only (CaM*A488) and double stained with anti-CaM*IgG-Alexa488/Aβ(1–42) HiLyte^TM^-Fluor 555 (CaM*A488/Aβ(1–42)). The results shown in (**B**) are the mean ± s.e. (*) *p* < 0.05 (i.e., statistically significant with respect to the control (CGN labeled with the Alexa488 FRET donor only)).

**Figure 4 ijms-22-01984-f004:**
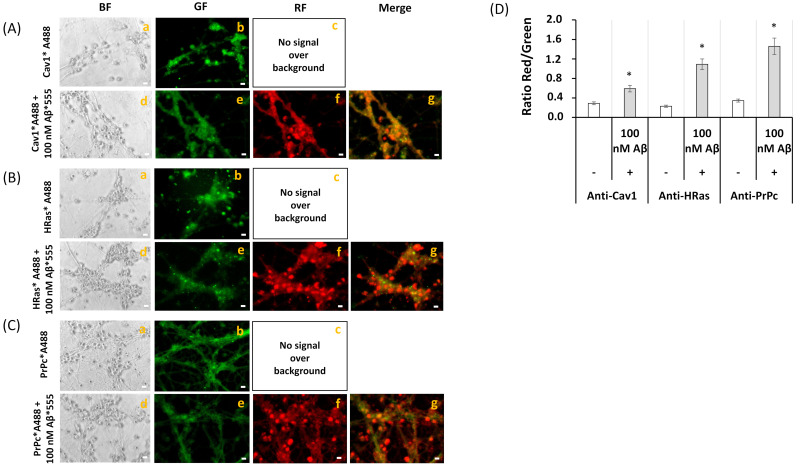
**Extensive FRET between the lipid rafts markers Cav1, HRas, PrPc, and Aβ(1–42).** (**A**) Representative quantitative fluorescence microscopy images of CGN stained with anti-Cav1 antibody (sc-894)/IgG-Alexa488 (Cav1*A488, a–c) or with anti-Cav1/IgG-Alexa488 and Aβ(1–42) HiLyte^TM^-Fluor 555 (Cav1*A488/Aβ(1–42), d-g). (**B**) Representative quantitative fluorescence microscopy images of CGN stained with anti-HRas antibody (sc-32026)/IgG-Alexa488 (HRas*A488, a–c) or with anti-HRas/IgG-Alexa488 and Aβ(1–42) HiLyte^TM^-Fluor 555 (HRas*A488/Aβ(1–42), d-g). (**C**) Representative quantitative fluorescence microscopy images of CGN stained with anti-PrPc antibody (7500997)/IgG-Alexa488 (PrPc*A488, a–c) or with anti-PrPc/IgG-Alexa488 and Aβ(1–42) HiLyte^TM^-Fluor 555 (PrPc*A488/Aβ(1–42), d-g). Bright-field (BF), green fluorescence (GF), and red fluorescence (RF) images are shown for each of the selected fields, and the orange-yellow areas (merge image) point out the higher intensity FRET regions (g, (**A**–**C**)). The exposure time for GF and RF images was 0.4 s. Scale bar inserted in fluorescence microscopy images = 10 μm. (**D**) Ratio of red/green fluorescence intensity per pixel (RF/GF) obtained from the analysis of fluorescence intensity data of CGN somas stained with (i) anti-Cav1/IgG-Alexa488 only (Cav1*A488) and double stained with anti-Cav1*IgG-Alexa488/Aβ(1–42) HiLyte^TM^-Fluor 555 (Cav1*A488/Aβ(1–42)); (ii) anti-HRas/IgG-Alexa488 only (HRas*A488) and double stained with anti-HRas*IgG-Alexa488/Aβ(1–42) HiLyte^TM^-Fluor 555 (HRas*A488/Aβ(1–42)); and (iii) anti-PrPc/IgG-Alexa488 only (PrPc*A488) and double stained with anti-PrPc*IgG-Alexa488/Aβ(1–42) HiLyte^TM^-Fluor 555 (PrPc*A488/Aβ(1–42)). The results shown in panel D are the mean ± s.e. (*) *p* < 0.05, (i.e., statistically significant with respect to the control, CGN labeled with the Alexa488 FRET donor only).

**Figure 5 ijms-22-01984-f005:**
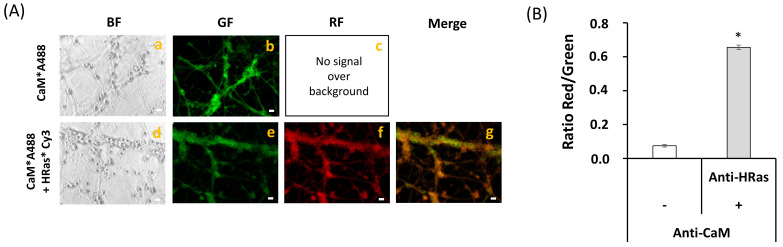
**Extensive FRET between CaM and HRas.** (**A**) Representative quantitative fluorescence microscopy images of CGN stained with the anti-CaM antibody (1716-1)/IgG-Alexa488 (CaM*A488, a–c) or with anti-CaM/IgG-Alexa488 and Anti-HRas antibody (sc32026)/IgG-Cy3 (CaM*A488/HRas*Cy3, d–g). Bright-field (BF), green fluorescence (GF), and red fluorescence (RF) images are shown for each of the selected fields, and the orange-yellow areas (Merge image) point out the higher intensity FRET regions (g). The exposure time for GF and RF images was 0.05 s. Scale bar inserted in fluorescence microscopy images = 10 μm. (**B**) Ratio of red/green fluorescence intensity per pixel (RF/GF) obtained from the analysis of fluorescence intensity data of CGN somas stained with anti-CaM/IgG-Alexa488 only (CaM*A488) and double stained with anti-CaM*IgG-Alexa488/anti-HRas*IgG-Cy3 (CaM*A488/ HRas*Cy3). The results shown in (**B**) are the mean ± s.e. (*) *p* < 0.05 (i.e., statistically significant with respect to the control, CGN labeled with the Alexa488 FRET donor only).

**Figure 6 ijms-22-01984-f006:**
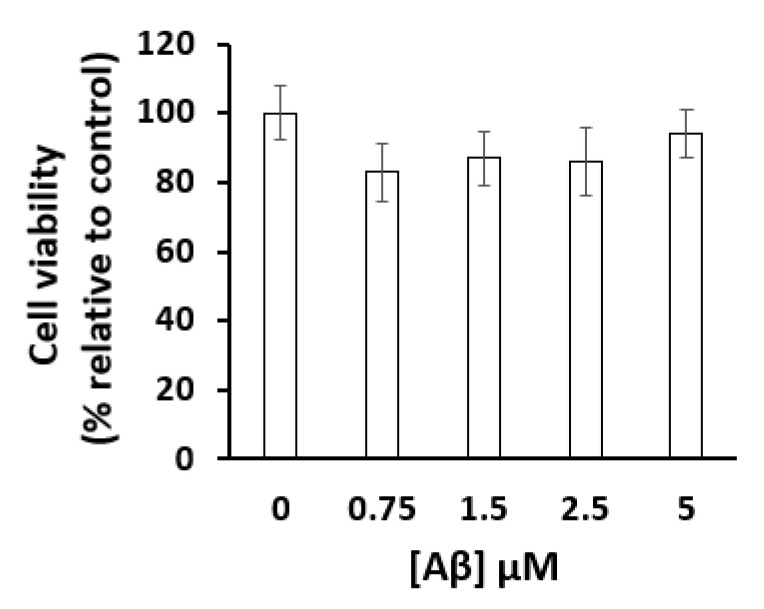
The cell viability of mature CGN was not significantly affected by Aβ(1–42) at the concentrations tested (0.75–5 µM) after 48 h of incubation at 37 °C and 5%CO_2_. Cell viability was measured using the 3-(4,5-dimethylthiazol-2-yl)-2,5-diphenyltetrazolium bromide (MTT) assay as indicated in the Materials and Methods section. The results were the average ± s.e. of experiments done in triplicate, with two different preparations of CGN.

**Figure 7 ijms-22-01984-f007:**
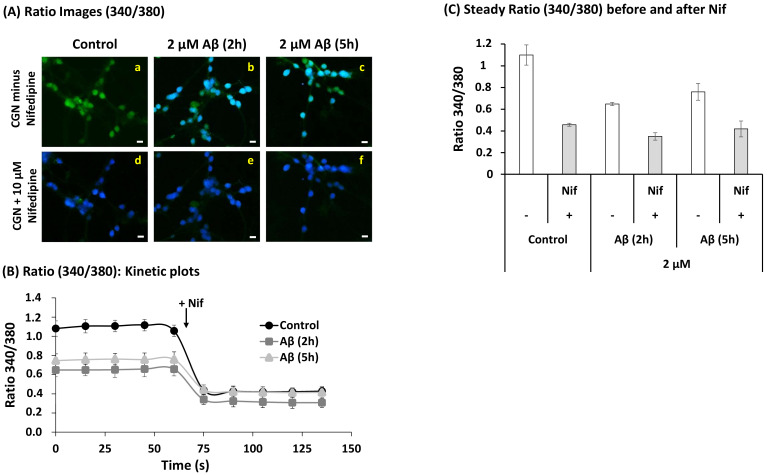
**Intracellular cytosolic calcium concentration decreases in the neuronal soma after Aβ(1–42) treatment.** Mature CGN were pre-incubated with 2 µM Aβ(1–42) for 2 h or 5 h at 37 °C and 5% CO_2_ and loaded with 5 µM Fura-2 for 60 min, as previously described in Section 2. Nifedipine (10 µM), a calcium channel blocker, was used as positive control to confirm the effect of the total blockade of LTCCs. (**A**) Representative ratio (340/380) images of Fura-2-loaded CGN of untreated (a) and treated cells with 2 µM Aβ(1–42) for 2 h (b) or 5 h (c) and after the addition of nifedipine in the control group (d) and in cells treated with 2 µM Aβ(1–42) for 2 h (e) or 5 h (f). Scale bar inserted in fluorescence microscopy images = 10 μm. (**B**) Kinetic plots of the average ± s.e. fluorescence ratio (340/380) before and after the addition 10 μM nifedipine (+Nif) at the time indicated by the arrow. (**C**) The analysis of fluorescence intensity data for GCN loaded with Fura-2 showed a statistically significant decrease (≈50%) in the steady ratio (340/380) after Aβ(1–42) treatment (2 h and 5 h) with respect to the control (untreated cells), *p* < 0.05. The addition of nifedipine totally blocks the entry of Ca^2+^ in the soma in untreated and treated cells as demonstrated by the decrease in the ratio showing that Aβ partially attenuates LTCC function. The ratio (340/380) values shown are the average ± s.e. of experiments done in triplicate with at least two different preparations of CGN (*n* > 400 neuronal soma of fields taken from at least six plates for each condition).

**Figure 8 ijms-22-01984-f008:**
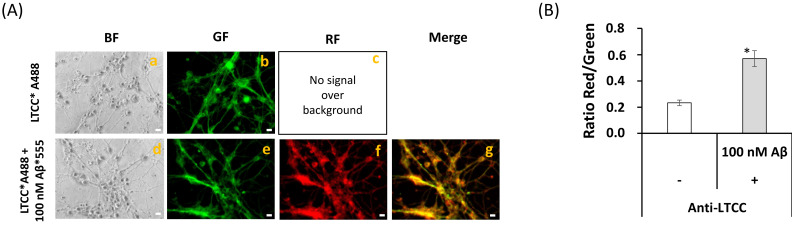
**Extensive FRET between LTTC and Aβ(1–42)**. (**A**) Representative quantitative fluorescence microscopy images of CGN stained with the anti-LTCC subunit α1C (sc-25686)/IgG-Alexa488 antibodies (LTCC*A488, a–c) or with anti-LTCC/IgG-Alexa488 and Aβ(1–42) HiLyte^TM^ Fluor 555 (LTCC*A488/Aβ(1–42), d–g). Bright-field (BF), green fluorescence (GF), and red fluorescence (RF) images are shown for each of the selected fields, and the orange-yellow areas (Merge image) pointed out the higher intensity FRET regions (g). The exposure time for GF and RF images was 0.4 s. Scale bar inserted in fluorescence microscopy images = 10 μm. (**B**) Ratio of red/green fluorescence intensity per pixel (RF/GF) obtained from the analysis of fluorescence intensity data of CGN somas stained with anti-LTCC/IgG-Alexa488 only (LTCC*A488) and double stained with anti-LTCC*IgG-Alexa488/Aβ(1–42) HiLyte^TM^ Fluor 555 (LTCC*A488/Aβ(1–42)). The results shown in (**B**) are the mean ± s.e. (*) *p* < 0.05 (i.e., statistically significant with respect to the control, CGN labeled with the Alexa488 FRET donor only).

**Figure 9 ijms-22-01984-f009:**
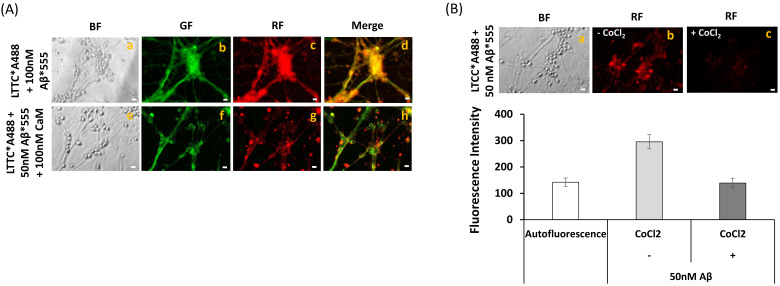
**Reversion by CaM of FRET between anti-LTCC/IgG-Alexa488 and Aβ(1–42)-HiLyte^TM^-Fluor555 (A) and quenching by Co^2+^ of the fluorescence of Aβ(1–42)-HiLyte^TM^-Fluor 555 (B)**. (**A**) Representative quantitative fluorescence microscopy images of CGN stained with anti-LTCC subunit α1C/IgG-Alexa488 and 50 nM of Aβ(1–42)-HiLyte^TM^-Fluor 555 (LTCC*A488/Aβ*555), after GF and RF image acquisition, 100 nM of purified CaM was added to the medium, and the plate was gently stirred for 15 min before acquisition of the images of selected fields in the presence of calmodulin. Bright-field (BF), green fluorescence (GF), and red fluorescence (RF) images are shown for the selected fields, and the orange-yellow areas (merge image) point out the higher intensity FRET regions. The exposure time for GF and RF images was 0.5 s. Scale bar inserted in fluorescence microscopy images = 10 μm. (**B**) Representative quantitative red fluorescence microscopy images of CGN stained with LTCC*A488/Aβ*555 in the absence and in the presence of 5 mM Co^2+^. After acquisition of RF images in the absence of Co^2+^, 5 mM of CoCl_2_ was added to the medium and images of the same field were acquired 5 min after the addition of 5 mM Co^2+^. The average intensity per pixel in the neuronal somas of five different fields were measured using HCImage software (*n* > 50 somas for each experimental condition), and are plotted below the fluorescence microscopy images of this panel (**B**). The results showed that 5 mM Co^2+^ completely quenched the fluorescence of Aβ(1–42)-HiLyte^TM^-Fluor 555, because the difference between the average intensity of the autofluorescence and of the fluorescence in the presence of 5 mM Co^2+^ was not statistically significant (*p* > 0.05).

## Data Availability

Data are available from our laboratory archives.

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
