# Peer review of "Binding of Amyloid β(1–42)-Calmodulin Complexes to Plasma Membrane Lipid Rafts in Cerebellar Granule Neurons Alters Resting Cytosolic Calcium Homeostasis"

_ijms, 2021, doi:10.3390/ijms22041984_

Round 1
Reviewer 1 Report
This is a weel desingned manuscript that claims that association of beta amiloid and CaM as a mechanism of Ca2+ alternations in neurons. Data sound strong and contribute to the field.
Author Response
We deeply appreciate your very positive and encouraging comments.
Reviewer 2 Report
This is an interesting and novel study. However, there are some minor concerns that should be addressed before publication.
Title:
I suggest the title “Binding of Amyloid β(1-42)-Calmodulin Complexes to Plasma Membrane Lipid Rafts in Cerebellar Granule Neurons Alters Resting Cytosolic Calcium” to be changed to “Binding of Amyloid β(1-42)-Calmodulin Complexes to Plasma Membrane Lipid Rafts in Cerebellar Granule Neurons Alters the Resting Cytosolic Calcium Homeostasis”
1) The English grammar of the manuscript is a bit poor, especially the abstract and in the introduction section. I strongly suggest a professional editing service to correct the language in the manuscript.
2) Line 11-13. The text needs to be corrected for grammatical error. “… in neurons because bind exogenous…”
3) Line 16-18. The sentence: “In this work, to experimentally assess the Aβ(1-42) binding capacity by intracellular CaM we have used primary cultures of mature cerebellar granule neurons (CGN) as a neuronal model” should be changed to: “In this work, to experimentally assess the Aβ(1-42) binding capacity to intracellular CaM, we have used primary cultures of mature cerebellar granule neurons (CGN) as a neuronal model”
4) In figure 3C one of the pictures is labelled “PRPc*A488 + 100 nM Aβ rojo” – what is the meaning of rojo?
5) In the legend of figure 7 the graph of Ratio 340/380 vs Time (s) is not explained
The main message in this study is that Aβ dimers can bind and inhibit LTCCs and thereby decreasing the resting cytosolic calcium concentration in CGN. However, in the discussion there is a lack of how this result is in harmony with other studies of how the neuronal calcium homeostasis is affected in Alzheimer’s disease.
Author Response
This is an interesting and novel study. However, there are some minor concerns that should be addressed before publication.
Answer: We deeply appreciate your positive comments and we have addressed all your minor concerns, see below.
Title:
I suggest the title “Binding of Amyloid β(1-42)-Calmodulin Complexes to Plasma Membrane Lipid Rafts in Cerebellar Granule Neurons Alters Resting Cytosolic Calcium” to be changed to “Binding of Amyloid β(1-42)-Calmodulin Complexes to Plasma Membrane Lipid Rafts in Cerebellar Granule Neurons Alters the Resting Cytosolic Calcium Homeostasis”
Answer: This suggestion has been accepted.
1) The English grammar of the manuscript is a bit poor, especially the abstract and in the introduction section. I strongly suggest a professional editing service to correct the language in the manuscript.
Answer: We have reviewed carefully the English grammar of the manuscript and introduced many typo and small grammar corrections in the revised manuscript. We have been given a very short of time for re-submission to look for a professional editing service for this task. However, on the basis of our large background with English publications we think that only minor grammar changes will be needed in the editorial processing of this manuscript. Only major corrections that led to phrase changes are highlighted in red color.
2) Line 11-13. The text needs to be corrected for grammatical error. “… in neurons because bind exogenous…”
Answer: Corrected, this phrase has been modified for the shake of clarity.
3) Line 16-18. The sentence: “In this work, to experimentally assess the Aβ(1-42) binding capacity by intracellular CaM we have used primary cultures of mature cerebellar granule neurons (CGN) as a neuronal model” should be changed to: “In this work, to experimentally assess the Aβ(1-42) binding capacity to intracellular CaM, we have used primary cultures of mature cerebellar granule neurons (CGN) as a neuronal model”
Answer: Corrected as suggested.
4) In figure 3C one of the pictures is labelled “PRPc*A488 + 100 nM Aβ rojo” – what is the meaning of rojo?
Answer: This comment refers to the Figure 4C, not to the Figure 3C. We thank you for letting us to be aware of this mistake. Rojo is red in Spanish, the fluorescence color of amyloid β(1-42) HiLyteTM-Fluor555, and we have corrected the labelling of the Figure 4C to normalize it with the labelling used in other panels of this Figure 4.
5) In the legend of figure 7 the graph of Ratio 340/380 vs Time (s) is not explained
Answer: It is explained in the revised manuscript. Thanks again for this observation.
The main message in this study is that Aβ dimers can bind and inhibit LTCCs and thereby decreasing the resting cytosolic calcium concentration in CGN. However, in the discussion there is a lack of how this result is in harmony with other studies of how the neuronal calcium homeostasis is affected in Alzheimer’s disease.
Answer: We have introduced two new phrases in the discussion of the revised manuscript to deal with this scientifically relevant comment. These phrases are highlighted in red color in page 13, lines 487-489 and 499-507. The calcium dysregulation induced by amyloid beta peptides in different neuronal cultures has been reported using in many cases exposure times to these peptides much longer than the 2 hours used in this work, which in turn leads to an intracellular concentration of free amyloid beta peptides much higher than that attained in our work. Also, it is to be noted that long times of exposure to amyloid beta can impair intracellular calcium homeostasis in neurons indirectly, due to the known pro-oxidant effects of amyloid beta peptides in the cells and the high sensibility to modulation by reactive oxygen species of several of the major calcium transport systems involved in neuronal homeostasis (reviewed in reference 28 of the manuscript, see also Hidalgo and Donoso - Crosstalk between calcium and redox signalling: from molecular mechanisms to health implications. Antioxid Redox Signal 2008;10: 1275-1312). In addition, the expression levels of major calcium signaling systems are markedly different in different type of neurons, not only of neurotransmitter’s plasma membrane receptors but also of endoplasmic reticulum calcium release channels. We avoided to comment on this and other related points in our previous submission, in order to minimize as much as possible speculative assertions in the Discussion.